# Innovative Strategy for Truly Reversible Capture of Polluting Gases—Application to Carbon Dioxide

**DOI:** 10.3390/ijms242216463

**Published:** 2023-11-17

**Authors:** Abdelkrim Azzouz, René Roy

**Affiliations:** 1Nanoqam, Department of Chemistry, University of Quebec at Montreal, Montreal, QC H3C 3P8, Canada; roy.rene@uqam.ca; 2École de Technologie Supérieure, Montreal, QC H3C 1K3, Canada; 3Glycosciences and Nanomaterials Laboratory, Department of Chemistry, University of Quebec at Montreal, Montreal, QC H3C 3P8, Canada; 4Weihai CY Dendrimer Technology Co., Ltd., No. 369-13, Caomiaozi Town, Lingang District, Weihai 264211, China

**Keywords:** CO_2_ capture, adsorbents, amines, polyols, dendrimers, zero-valent metals

## Abstract

This paper consists of a deep analysis and data comparison of the main strategies undertaken for achieving truly reversible capture of carbon dioxide involving optimized gas uptakes while affording weakest retention strength. So far, most strategies failed because the estimated amount of CO_2_ produced by equivalent energy was higher than that captured. A more viable and sustainable approach in the present context of a persistent fossil fuel-dependent economy should be based on a judicious compromise between effective CO_2_ capture with lowest energy for adsorbent regeneration. The most relevant example is that of so-called promising technologies based on amino adsorbents which unavoidably require thermal regeneration. In contrast, OH-functionalized adsorbents barely reach satisfactory CO_2_ uptakes but act as breathing surfaces affording easy gas release even under ambient conditions or in CO_2_-free atmospheres. Between these two opposite approaches, there should exist smart approaches to tailor CO_2_ retention strength even at the expense of the gas uptake. Among these, incorporation of zero-valent metal and/or OH-enriched amines or amine-enriched polyol species are probably the most promising. The main findings provided by the literature are herein deeply and systematically analysed for highlighting the main criteria that allow for designing ideal CO_2_ adsorbent properties.

## 1. Introduction—Air Pollutants and Environmental Impacts

Air pollution has a great negative environmental impact on natural cycle equilibria, biodiversity, and human health. Air pollution should be restricted to physicochemical processes triggered by human activities, excluding similar processes produced by natural phenomena, if one assumes that nature is capable of self-regenerating. Indeed, nature already turned out to be capable of readapting by establishing new equilibrated cycles of elements and living species after major events that produced great changes in Earth evolution.

Air pollution has often been tackled by attempts to remove a given pollutant downstream from its release in the atmosphere. This approach was already doomed to failure because it does not take account of the matter and energy inputs and outputs in the environment, including the intakes/losses from/towards the extraterrestrial space and in-ground accumulation and storage (Figure 1). In other words, a more sustainable strategy should consider all the interactions occurring between all natural cycles of elements and energy involved in pollutant production and its interaction with the surrounding environment. 

Any air pollution should be regarded as a major perturbation of the entire environment as a whole. Air pollution may arise from solid particles, liquids (acid rains, organic sprays…) and gas compounds (S and N oxides, volatile organic compounds…). Thus, air pollutants can be solid particles, commonly denoted as suspended particulate matter (SPM), liquid droplets, volatile species or gases such as oxides of carbon (CO_x_, mainly CO and CO_2_), sulphur (SO_x_, mainly SO_2_ and SO_3_), and their combination (COS), nitrogen (NOx, mainly NO, NO_2_, N_2_O_3_, N_2_O_3_…), volatile organic compounds, and others [1]. 

Among air pollutants, there is also a wide variety of volatile organic compounds (VOC). These include not only methane and other gases, volatile hydrocarbons arising from incomplete fuel combustion, chloro-fluoro-carbons (CFCs) and pesticides, but also household and cosmetic products. They are carbon-based compounds displaying vapor pressure of at least 0.01 kPa at 25 °C and thorough volatility under ambient conditions [2]. Even in very low concentrations, their mere occurrence in the atmosphere can induce marked adverse effects on ecosystems and human beings [3]. VOCs are involved in the rise of ground level ozone that unavoidably accentuates VOC radical dissociation and contributes to the accentuation of the greenhouse effect, triggering smog processes with severe health issues [4].

Fossil fuel combustions and agricultural processes predominantly eject primary air pollutants such as CO_2_, sulphur oxides (SO_x_, mainly SO_2_ and SO_3_), and high amounts of nitrogen oxides (NOx, mainly nitrate, nitrite, and nitrogen protoxide). Secondary pollutants are not directly emitted, but rather generated by primary pollutant conversion via chemical processes such as the so-called photochemical smog. The contents and distribution of both primary and secondary pollutants in the atmosphere can be significantly influenced by temperature, pressure, humidity, and air mass convection. 

Ozone, CO_2,_ and acid rains (HNO_3_, (NH_4_)_2_SO_4_, H_2_SO_4_) mainly produced by NO_x_ and SO_x_ have direct negative impacts on plants, waters, and soils for buffering capacity [5]. Excessive acid rain induces water and soil acidification affecting aquatic ecosystems and biodiversity equilibria. In high amounts, harmless CO_2_ becomes an air pollutant that contributes to the greenhouse effect of the atmosphere, inducing global climate changes [2]. 

Ideally, pollution prevention should not focus only on controlling major emissions of gas pollutants, but also on upstream contributors to the gas production. Many approaches have been tackled for gas pollutant capture, mainly through adsorption techniques. Nonetheless, such techniques can only be profitable when applied to sufficiently concentrated air pollutants in emission flues. Otherwise, upstream air pollutant concentration should be necessary.

## 2. Greenhouse Gases and Global Warming

A greenhouse effect is induced by a gas that absorbs, stores, and re-emits infrared radiation in the atmosphere. There already exists a natural greenhouse effect that preserves life on Earth. This effect involves balanced radiation exchanges between their production by the planet surface and by inelastic collisions of sun photons in the atmosphere and their loss in space [5]. The slight accumulation of decelerated sunlight photons results in atmosphere warming. This phenomenon can be accentuated by the presence of excessive amounts of other greenhouse gasses [6]. The primary greenhouse gases in the Earth’s atmosphere are water vapours, CO_2_, methane, and nitrous oxide. Water vapour is the most important natural greenhouse gas, being responsible for the natural greenhouse effect and for the main energy changes that lead to the formation of clouds and condensation turning into rains, snows, and ices.

Global warming may be a natural process produced by many causes including periodicity in the Earth’s evolution. However, there is no doubt that increasing greenhouse gas contents in the atmosphere is probably the most visible contribution to global warming. The concentration of gas pollutants in the atmosphere is still increasing because of the continuous human activity growth. This growth is beyond the environmental capacity to preserve equilibrium between the natural cycles of elements and living species. CO_2_ is one of these pollutants, being a major greenhouse gas (GHG) with probably the highest contribution to global warming and its negative aftermaths through ocean acidification [7] and CO_2_ exchange with the atmosphere and soils on flora and fauna [8]. Deforestation was found to accentuate these sequels by reducing the natural capacity of CO_2_ retention leading to uncontrolled increase of the greenhouse effect. 

## 3. CO_2_ Natural Cycle and Major Emission Sources

CO_2_ is produced by natural processes such as respiration and aerobic fermentation along with natural forest fires caused by the liquid droplet magnifying effect, volcano eruptions, electrical discharge by lightning, and other apparently ‘’spontaneous’’ phenomena. In the meantime, CO_2_ is mainly consumed by absorption mainly via photosynthesis, adsorption in soils, and dissolution (Figure 2). 

CO_2_ emission and consumption processes can be balanced by CO_2_ natural cycles only in interaction with the following: *i*. the most oxidized and most reduced forms in other cycles of elements; *ii*. vegetal and animal biomass production and storage by anaerobic formation of excess biomass in the forms of gases, oils, biochar, and coals [9].

Nevertheless, anthropogenic activities often involve combustions of coal or hydrocarbons, fermentation of sugars in industry, and livestock respiration. These anthropogenic processes involve an estimated growth in fossil fuel consumption of ca. 1.3% per year [10]. This results in CO_2_ excesses that are not necessarily assimilated by natural cycles in correlation with cycles of other elements and energy. Any perturbation of parts of this entire interconnected cycle scaffold can have a great influence on CO_2_ content in the atmosphere. The latter already reached ca. ~417 ppm due to global CO_2_ emissions around 40 gigatons/year this last decade and an almost constant growth of approximately 500 million metric tons/year since 1950 [11]. This content is one of the main factors for global warming, ocean acidification, melting of glaciers, and other negative aftermaths.

Nowadays, it is well established that CO_2_ emissions mainly arise from industries and transportation which unfortunately operate almost exclusively with fossil fuels [3,12,13,14,15]. CO_2_ emitted is released unilaterally into the atmosphere, a common space shared by the entire planet, but without sharing in productivity income and damage caused by pollution. Pollution by CO_2_ is a source of many negative impacts on air and water qualities and on animal and human health [12,13,14,15,16]. Ample literature is now available in this regard.

The main gaps and challenges regarding global warming reside in governance’s difficulty of prioritizing the CO_2_ environmental issue and implementing sufficient large-scale facilities for carbon dioxide removal (CDR) that include permanent CO_2_ storage [17]. Global warming mitigation requires simultaneous drastic CO_2_ emissions reduction and carbon removal intensification, combined with global and comprehensive policies regarding CO_2_ capture, storage, and valorisation that benefits all countries [18]. The so-called “net zero” objective until 2050, i.e., non-natural CO_2_ emissions lower or at most equal to CO_2_ capture and sequestration, is a utopia without a global concertation in the international community even at the expense of specific economic interests of industrialized countries [11].

CO_2_ capture from the atmosphere is already a lost race against unstoppable and increasing fossil energy consumption and gas emissions and more particularly without fairly sharing data and benefits. One of the primary targeted objectives in CO_2_ capture consists in the compensation of the excessive and fast depletion in fossil fuel sources [19]. Most industry research and development of CO_2_ capture technologies are focused on commercialization for their own economic interests. A successful approach in this direction should take into account the existing or already implemented renewable energy sources. Except for the biological CO_2_ conversion by vegetal photosynthesis after reforestation or injection in agricultural greenhouses, gas sequestration strategies in soils are questionable due to major gaps, more particularly when it targets solid storage via in-situ mineralization in underground geological layers without deep knowledge about the pre-sequestration physicochemical features of the underground host reservoir and possibility to predict the post-sequestration behaviour [18]. Science gaps reside in knowledge advancements mainly focused on developing new adsorbents for CO_2_ capture and catalyst materials for its conversion into added-value products. These approaches are only intended to at most compensate CO_2_ emissions but not to restore the carbon balance in the environment [11].

## 4. Strategies for CO_2_ Capture and Potential Valorisation Routes

Plants, soils, and oceans are natural ab/adsorbents of CO_2_ and are already acting as reservoirs of this gas. Large amounts of CO_2_ are stored in both physically and chemically ab/adsorbed forms, mainly as metal carbonates [16]. Deep knowledge about the physicochemical behaviour of these ab/adsorbing host matrices allows for envisaging approaches that mimic natural CO_2_ capture. For tackling this issue, scientists have triggered extensive research, more particularly after 2000, inasmuch as between the period 2001–2021, 18,500 publications on CO_2_ capture and sequestration have been reported, with an additional 4500 in the last two years [11].

Nonetheless, given the lack of a comprehensive sustainable strategy for fossil energy sources, CO_2_ capture, sequestration, and supposedly ‘’green’’ recycling processes into added-value products have become the only approaches to address this detrimental environmental issue. CO_2_ capture can be achieved at different stages in the production-emission chain of this gas, namely before and after fossil fuel combustion or once already released in the atmosphere (Table 1).

A Japanese team developed probably the most efficient CO_2_ capture, called IPDA, affording almost 99 removal yield from air (400 ppm CO_2_) [20]. This was achieved using isophorone diamine (IPDA; 3-(aminomethyl)-3,5,5-trimethylcyclohexylamine; CH₃)₃C₆H₇(NH_2_)(CH_2_NH_2_)), a colourless liquid diamine, usually employed as a precursor in polymer synthesis and coatings. The process consists in CO_2_ absorption in liquid IPDA dissolved in water which results in the formation of a solid carbamic acid, affording a 201 mmol CO_2_/h for each mol IPDA in a CO_2_/IPDA ratio higher than unity and a 100-h stability during the DAC process (Figure 1). Complete CO_2_ desorption can be achieved without IPDA degradation even after repeated adsorption desorption cycles for 100 h.

After heat recovery, if any, for various purposes including domestic or agricultural greenhouse heating, more particularly in cold countries, the capture and concentration of CO₂ with or without purification are intended either for permanent geological sequestration in underground layers, or for direct use in these agricultural greenhouses, or conversion to value-added carbon-based compounds. Pre-combustion CO_2_ capture consists in extracting the gas from fossil fuels (15–50% CO_2_ in this mixture) before complete combustion. This method is used for high pressure thermal coal gasification into synthesis gas (Syngas, a CO/H_2_ mixture). This is achieved through partial oxidation under steam and oxygen/air stream. Post combustion CO_2_ capture is a low-pressure separation from gas emissions (ca. 5–15% CO_2_) after fossil fuel combustion in air, unlike oxy-fuel combustion capture which uses nearly pure oxygen and recycled flue gas.

Except for agricultural use in greenhouses, these approaches are neither viable nor sustainable. This is due to the fact that the concentrations of CO_2_ in the main emission sources are paradoxically low for profitable technologies as compared to the critical CO_2_ levels attained so far in the atmosphere. As long as the world’s economy is still using fossil energy sources, direct CO_2_ capture from flue emissions is doomed to failure but still remains a necessary step, because in most cases it involves more CO_2_ production than retention. However, this step can only be justified for sufficient CO_2_ concentration regardless of the consecutive purpose targeted.

In other words, CO_2_ concentrations are sufficiently high to induce negative impacts on the environment, but in the meantime sufficiently weak not to justify direct gas sequestration in soil, unless CO_2_ is previously concentrated. Thus, CO_2_ is particularly suitable for underground sequestration, more particularly in basic rocks rich in metal hydroxides, being heavier than air with a density of 1.977 g/L. This approach can be viable as long as this technique does not produce additional CO_2_, and if both geological and environmental risks are fully evaluated. For instance, CO_2_ injection in ground waters was already found to cause loss of wells, pumps and pipes corrosion, and other problems [16].

So far, many techniques have been tested for reducing greenhouse effects. Some of these were based on cryogenic methods [7], retention and conversion by microbial and algal systems [21], membranes separation [9,22], and others. Cryogenic capture involves direct liquefaction of high purity and water-free CO_2_ to avoid pipe clogs. This method also requires multiple compression and cooling stages to selectively liquefy CO_2_ from other gases. However, it is worth mentioning that the energy consumption (0.6–1.0 kWh/kg CO_2_ captured) is about twice higher than that of the chemical absorption [9,21,23,24]. Membranes displaying 1500–3000 m^2^ exchange surface per m^3^ of contactor can be used in CO_2_ multistage separation in flue gases, but the process imposes the use of polymeric membranes for low temperatures and ceramic counterparts for high temperatures. Reportedly, polymer membranes showed higher performances than their ceramic-based counterparts, but much less thermal stability when exposed to heat [9,23].

More effective techniques for potential CO_2_ conversion into added-value products are undoubtedly those involving absorption in base-containing liquids [25]. In this regard, adsorption on basic metal oxides such as magnesium oxide (MgO) or calcium oxide (CaO) to form stable carbonates [26] is possible, but their decomposition for further CO_2_ reuse requires heating. Moreover, chemical retention of CO_2_ by base-like liquids or solids produces undesirable wastes and could not be regarded as being sustainable alternatives. The reversible capture of CO_2_ appears to be a more profitable route, more particularly when achieved on microporous adsorbents displaying high specific surface area and affinity towards CO_2_.

## 5. CO_2_ Absorption Methods

CO_2_ is a colourless, non-flammable, and non-toxic gas that can be breathed by humans and animals. However, it can cause discomfort and nasal irritation at 2–5% concentrations in air, asphyxia when in excess in unventilated enclosures, and can even be lethal at 10% concentrations [16]. However, the mere presence of moisture or liquid water may significantly reduce these risks, due to the appreciable solubility of CO_2_. In the meantime, such solubility in waters can also be detrimental, because it unavoidably leads to excessive acidification of aqueous media that affects biodiversity [16].

CO_2_ acts as a Lewis acidic gas due to the low electron density on its central carbon atom and peripherical oxygen atoms that exhibit non-bonded electron pairs. This last property is a key argument for applying ab/adsorptive methods inasmuch as it confers the following: *i.* affinity towards basic sites of metal oxides, amines, electron pair acceptors, and polar molecules; and *ii.* capacity to favour the rise of H-bridges with hydrogen-rich chemical structures such as water and carbonate-like associations. This also explains CO_2_ solubility in oxygenated organic solvents such as alcohols, glycols, ethers, and ketones, more particularly in acetone and ethanol [27]. CO_2_ is also known to display strong affinity towards basic aqueous solutions of NaOH, KOH, their mixtures, and other bases. Ionic liquids have also been tested in CO_2_ capture based on the local polarity of the C=O bonds. Nevertheless, such liquids are not common in nature and often require sophisticated preparation techniques and severe constraints in handling. Moreover, CO_2_ solubility was found to mainly involve chemical absorption [28,29]. 

The use of liquid amines is probably the most common method in terms of CO_2_ uptakes, but the unavoidable need for thermal regeneration constitutes a major drawback [29]. Moreover, the very use of liquid amines for absorbing CO_2_ from flue gases can detrimentally induce amine evaporation and even degradation, even with the combined use of membranes with solvents. Primary alkanolamines such as monoethanolamine (MEA , HOCH_2_CH_2_NH_2_), secondary (diethanolamine) , and tertiary counterparts (triethanolamine) [30,31], along with amidine and guanidine , were already found to easily react with CO_2_ through an exothermic reaction that generates carbamate species [32]. The intensity of this reaction varies according to the molecular structure of these amines, in addition to the need for thermal regeneration which still remains a major and common shortcoming. 

## 6. CO_2_ Adsorption on Solids

CO_2_ was already found to readily adsorb on a variety of solid surfaces including microporous and mesoporous aluminosilicates (zeolites and cationic clay minerals), structured carbons, metal oxides, hydrotalcite, and modified micro-/mesoporous materials. Depending on CO_2_ interaction with the solid surface, adsorption can involve chemosorption and/or physisorption. Chemical adsorption involves CO_2_ capture via covalent chemical bonding on solid materials and rearrangement of the electron density. Such processes usually take place at temperatures above 200 °C and are essentially irreversible. Anionic and basic clays also turned out to be suitable as CO_2_ adsorbents not only for precombustion processes at temperatures below 200 °C [33,34] but even under ambient conditions [35,36].

As compared to liquid host-matrices, solid adsorbents have significant advantages for energy efficiency reasons. The incorporation of amine into clay minerals, zeolites, or other mixed oxides is supposed to facilitate the adsorbent handling and regeneration without amine loss. This should reduce contamination risks by amine release and/or decomposition and is assumed to improve the thermal and mechanical stability, the dispersion of active sites, and surface reactivity towards CO_2_. Reportedly, purely physical insertion of amines on solid supports such as activated carbon produced higher CO_2_ uptake at relatively high pressures [24]. CO_2_ instantly adsorbs on supported bases and more particularly supported amine under ambient conditions, but desorption for adsorbent regeneration requires heating. The use of low-cost wastes as heat sources is a partial solution, which only minimizes the negative environmental impacts of such an approach for CO_2_ capture.

Researchers tested new technologies for CO_2_ capture from air pollution by supported oxygenated organic compounds like metal organic frameworks (MOFs). As compared to microporous materials (zeolites, activated carbons), MOFs display higher pore volume and surface area that favour adsorption of CO_2_. They consist of organic bridging ligands to metal to form three-dimensional extended correlations with uniform pore size and one or more metal ions (e.g., Al^3+^, Cr^3+^, Cu^2+^, or Zn^2+^) chelated by carboxylate and/or pyridyl groups. In spite of their high surface-to-bulk ratio, their unique structural properties, robustness, high thermal and chemical stabilities, and the use of materials prepared through sophisticated techniques such as MOFs is difficult to justify [37]. This is supported by the fact that much lower-cost and more available materials like zeolites can allow for affording higher CO_2_ uptakes at lower pressures [37,38]. It is worth mentioning that the effect of pressure cannot be dissociated from that of temperature, since increasing temperature imposes higher pressure for achieving higher CO_2_ retention capacity, more particularly on zeolites [39].

## 7. Design of Reversible CO_2_ Capture

When physically adsorbed, CO_2_ usually accumulates in non-stoichiometric amounts with respect to the adsorption site density. Depending on the retention strength, pressure/temperature swing adsorption cycles can be applied to remove and concentrate CO_2_ [21]. Such adsorbents can also operate through repetitive adsorption-desorption cycles, or act as separation membranes between two differently concentrated media.

CO_2_ displays acidic character and is expected to interact with liquid media or solid matrices exhibiting base properties. Amine-containing liquids or solid matrices have already shown satisfactory CO_2_ uptakes but through the unavoidable chemical formation of carbamate groups (Figure 3). This imposes required thermal regeneration as reported by a large literature on CO_2_ retention attempts by matrices displaying basicity. An increasing interest has also been devoted to less basic anionic clay minerals such as layered double hydroxides (LDH) and hydrotalcites, their natural counterpart. They could be convenient adsorbents for CO_2_ removal, but their regeneration still needs drastic thermal desorption. 

However, the concept of truly reversible CO_2_ capture for further gas concentration has scarcely been tackled. This concept is based on the attenuation of CO_2_ interaction strength with the absorbing/adsorbing solvents and host matrices during the gas capture and consecutive release. An essential requirement for a truly reversible retention of CO_2_ resides in the use of solids that combine a high number of adsorption sites but much weaker basicity as compared to amines. 

The rise of the new concept of “Truly Reversible Gas Capture” has stimulated research on crystalline aluminosilicates such as zeolites, cationic clay minerals, and volcanic tuffs. This interest is based on their weak Bronsted acidity on their silanols and sufficient Lewis basicity on their lattice oxygen atoms. Such materials appear to be suitable matrices not only for CO_2_ capture purposes but also for surface modification for adsorption capacity improvements. So far, successful attempts have been achieved through the synthesis of clay-OH-dendrimer composites affording CO_2_ retention capacity by far higher than the one-to-one stoichiometry of carbamate formation on conventional amine-based adsorbents. Organoclays obtained by the incorporation of polyol (or alcohol)-based dendrimers are assumed to act as “lungs” that can easily release the adsorbed gas even at room temperature or around (20–50 °C) in CO_2_-free media or under strong gas streams without need of thermal regeneration [35,40,41,42,43,44]. This concept has barely been examined up until today, but has already been extended to hydrogen capture and storage by low-cost sponge-like matrices with high surface-to-bulk ratios [45,46]. It can certainly be applied to the design of expanded matrices that can “respire” NO_x_, SO_x_, volatile organic compounds (VOC) and other molecules in the near future. This also opens new prospects for medical, agricultural, and industrial applications involving catalysis and adsorption.

In spite of their apparent surface basicity (Bronsted), the lattice oxygen atoms surrounding their exchange sites are capable of acting as adsorption sites for CO_2_. The exchangeable cations are the main factor that can modify the acid-base properties. Indeed, alkali metal cations showed higher affinity towards CO_2_ as compared to heavier metal ions whose high polarizing is known to dissociate solvating water molecules and Bronsted acidity.

However, these intrinsic acid-base properties do not confer sufficiently high adsorption capacity. Adequate modifications of the solid surface for raising the number of adsorption sites can be achieved by incorporating weakly basic to amphoteric compounds such as poly-alcohols. The incorporated hydroxyl groups are expected to display such a weak interaction that CO_2_ can even be released at room temperature in CO_2_-free media or by forced convection in a gas stream. Such an adsorption is assumed to take place via the formation of carbonate-like associations. Poly-alcohols in polymeric or dendritic forms can produce organo-zeolites or organo-clays with high effectiveness in the reversible CO_2_ capture. 

Clay minerals turned out to be interesting low-cost raw materials for the capture of pollutants dispersed in gas [35,40,41,42,43,44] or in liquid media [47,48]. Organo-clays obtained through montmorillonite intercalation with commercial polyol dendrimers (Boltorn^TM^ H-20 to H-40, Malmaö, Sweden) and soya-derived polyglycerol dendrimers already showed promising performances in the reversible CO_2_ capture. These performances were explained in terms of amphoteric character that promotes the formation of non-stoichiometric clusters of CO_2_ molecules weakly bound to each other around a single OH group. This concept of multilayer hydrogen adsorption around coordinated adsorption sites has already been tackled with attempts to use Metal-Organic Frameworks (MOF) [49]. 

## 8. Interactions on Clay-Supported Polyalcohols

Adsorption on polyalcohol-based solids could be regarded as a more suitable alternative for CO_2_ capture, as illustrated by CO_2_ uptake improvement upon alcohol incorporation [25,50]. However, it was already demonstrated that their major drawback resides in their compacted structure that arises from their intrinsically internal H-bridges. In other words, excessive hydroxyl density in the polyalcohol molecular structure is detrimental for ROH:CO_2_ interaction due to the occurrence of competitive inter and intramolecular OH:OH- bridges.

This is a major drawback that can be overcome by using solid supports capable of promoting hydroxyl interactions with the solid surface and consequently polyalcohol dispersion at the expense of those occurrence between the organic chains. Natural silicates and layered aluminosilicates such as clays minerals are convenient and adequate materials for such a purpose, displaying large surface area with terminal silanols [51]. The latter can interact with incorporated ROH when modified by intercalation resulting in ordered assemblies or well dispersed alternate inorganic-organic layers exhibiting combined properties from the two components.

On a clay surface for instance, multiple homo- and heteromolecular interactions can take place between CO_2_, moisture, alcohols, and a host-clay surface. Smectite-type clay minerals such as montmorillonites may easily be intercalated with polyalcohols, and water seems to play an essential role in clay surface interactions with both polyol molecules and CO_2_ (Figure 4).

Polyalcohol molecules are supposed to involve interaction of their hydroxyls with the clay surface. Monolayer incorporation of polyalcohol molecules between silicate sheets appears to induce higher stability than with high layer numbers. A possible explanation should consist in the amphoteric to slightly basic character of the hydroxyl groups. The latter are assumed to promote acid-base interaction preferably with more acidic out-of-plane silanol groups of the montmorillonite surface [52,53]. This should occur at the expense of other hydroxyls from the next neighbouring polyalcohol chains. A similar competitivity may take place with water molecules but given the appreciable affinity of CO_2_ and alcohols towards water, the latter is rather expected to promote synergy in improving CO_2_ capture.

Among the various types of clay materials such as smectites, kaolinites, palygorskites, sepiolites, and others, the first category is particularly interesting due to its relatively high silica content. The latter offers terminal silanol groups that can act as potential adsorption sites for water, alcohols, and even CO_2_ via H-bridges due to their hydrophilic character and slight acidity. Smectites include bentonites and their purified clay mineral forms of sodium and calcium montmorillonites which are widely used for diverse purposes [54]. Al^3+^ substitution for Si^4+^ in the tetrahedral sheet, or trivalent ions (Fe^3+^ or Al^3+^) for divalent (Fe^2+^ or Mg^2+^) in the octahedral sheets, and/or the addition of monovalent (K^+^ or Na^+^) or divalent (Mg^2+^ or Ca^2+^) cations in the interlayer space, can give rise to a wide variety of ion-exchanged smectites that differ not only by exchangeable cations but also through their silica/alumina ratio and clay sheet structures.

Bentonite purification through repetitive ion-exchange and settling steps in NaCl solution and/or slight short acid treatment in aqueous 0.1 N HCl and sometimes under ultrasound exposure allow for affording highly Na^+^-montmorillonite devoid of dense silica phases, carbonates, organic impurities, and other metal cations [25,41,42,43,55]. Clay pillaring is another route to obtain interesting support for organic moieties exhibiting affinity towards CO_2_ with rigid structure similar to those of zeolites [55,56].

Both purified bentonite and pillared clay materials can be prone to acidity attenuation or basicity improvement through the incorporation of organic moieties bearing adsorption sites with more or less affinity for CO_2_ capture. This can be achieved through physical adsorption (intercalation) or by chemical grafting including ion- exchange and pillaring by a variety of species.

Intercalation is a purely physical and reversible inclusion of different species between two clay sheets. This process depends on the geometrical, physical, and chemical characteristics of both the inserted species and clay surface. It is noteworthy that intercalation (physical insertion) of inorganic or organic species can often be the first step of a chemical process via ion exchange, covalent grafting, and pillaring. This is expected to involve a mixture of van der Waals, dipole-dipole, Lewis acid-base interactions, H-bridges, and electrostatic and electric charge interactions between the stacked layers [51,57,58]. Given their various modification procedures and multiple applications [58,59,60,61,62], intercalated clay minerals are promising materials with tailored properties according to the inserted species and its chemical functions.

## 9. Chemical Grafting

Chemical grafting occurs via the formation of covalent bonds usually between an organic moiety bearing an anchorable group and host-surface. Here also, a large variety of materials have been tested as solid supports for hosting chemically grafted species. Materials displaying expandable structures and high surface-to-bulk ratios such as metal-organic-frameworks, nanostructured carbons or meso-and microporous silicas have particularly attracted interest [49,63,64,65,66,67,68,69]. However, their low chemical and thermal stability, complex synthesis procedures, and high preparation costs are major obstacles for their implementation in manufacturing plants and industrial applications. 

Various organic moieties such as amines, polymers, polyols, dendrimers, and others can be grafted on clay surfaces [70,71]. High surface areas bearing a larger number of grafted amines is an essential requirement for achieving high CO_2_ uptake. The type of grafted amines (primary, secondary, or even tertiary) will determine the amount of CO_2_ adsorbed and the energy required for its release and for adsorbent regeneration. Amines are weak bases which, however, impose sufficiently strong acid-base interaction with a binding energy in the range of 50–100 kJ/mol depending on the species formed by CO_2_ adsorption on the amine [72,73].

Dendrimers are interesting organic moieties for chemical grafting on solid surfaces. A wide variety of dendrimers have been synthesized these last three decades, and an ample literature has been reported in this regard [74,75,76], starting from the simplest structures [77] to progressively bulkier and more scaffolded structures [78,79,80,81,82,83,84,85,86,87,88,89,90]. Their intrinsic properties [83,91,92,93] are designed to confer specific features to the hybrid materials when supported on inorganic supports according to the applications targeted [35,41,42,43,44,50,94,95,96,97,98,99,100,101,102,103,104].

Dendrimers are radial assemblies of polymeric but monodispersed macromolecules with repetitive and more or less circular sequences of monomer layers bounded around a core [105]. Dendrimers have a large number of reactive surface groups that can act as anchorable sites for more peripherical moiety layers. Amino and hydroxyl terminal groups are particularly interesting not only for CO_2_ adsorption but also for grafting additional layers with higher numbers of terminal chemical functions. One of the most tested dendrimers in CO_2_ capture is probably PAMAM (poly(amidoamine), synthesized starting from ammonia or ethylenediamine scaffolds followed by grafting of polypropylenimine dendrimers (PPI) [106]. Reportedly, some dendrimers have also been synthesized through direct grafting on a mesoporous inorganic core such as a MCM-41 crystallite [107,108] and/or SBA-15-like silica [109]. The use of mesoporous materials with larger pore size opens promising prospects for the incorporation of higher generation and bulkier dendrimers [105,106,107]. Nonetheless, care should be taken in direction because of the rise of detrimental intra-dendrimer structure collapse and loss in porosity and adsorption capacity upon internal interactions between the different chemical groups of neighbouring dendritic branches.

On clay minerals, chemical grafting of such organic moieties confers hydrophobicity to the inorganic surface. On smectites, silanols are suitable sites for chemical grafting through silylation of anchorable groups such alkoxy-silane or alkoxy-siloxane and release of water and/or alcohol molecules (Figure 5 and Figure 6). This often results in increased interlayer spacing and structure expansion with favourable higher surface area, porosity, and pore volumes as compared to the starting clay mineral. These changes greatly improve the adsorption capacity and affinity towards CO_2_ when the grafted organic moiety bears adequate chemical functions.

Polyols dendrimer grafting gives rise to new regenerable materials with weak basic hydroxyl groups but high amounts of CO_2_ weakly retained [44]. 2,2-Bis(hydroxymethyl)propionic acid (bis-MPA), trimethylolpropane (TMP), pentaerythritol and its homolog ethoxylated pentaerythritol (PP50) are common scaffolds for polyhydroxylated dendrimers such as the Boltorn^TM^ HX family (where X can be 20, 30, 40, 50…) with 2nd, 3rd, 4th, 5th… generation are some examples of typical and commercially available polyol dendrimers. The number of their hydroxyl groups governs the strength of H-bridge interactions occurring with surrounding molecules [110,111]. Here also, the number generation is an essential requirement for increasing the number of terminal groups, but not necessarily for improving the adsorptive properties, inasmuch as excessively scaffolded structures often undergo collapse due to internal interaction between the chemical groups of the dendritic branches. 

Already synthesized dendrimers can also be directly grafted as such on aluminosilicate surfaces through silylation processes (Figure 7). Such composites showed high physical adsorption capacities in molecular hydrogen storage attempts [112,113,114,115,116]. Promising performances reported for a large variety of nanoporous carbons, wood-based activated carbons [117], graphite, nanostructured carbons, and carbon aerogels [63] still remain to be confirmed for envisaging potential applications for CO_2_ capture.

Scaffolded structures like 4-(triethoxysilyl)aniline or 3-(triethoxysilyl)propylamine can be anchored on the silanol groups via their (EtO)_3_Si-R groups by mere heating with ethanol elimination [118]. The synthesis of anchorable dendrimers and the in-situ anchoring of dendrimers on already grafted -Si-O-Si- bridges resulted in dendrimer-based organoclays that may act as precursors for more complex scaffolds (Figure 8). Dendrimers and dendrons may be converted on the clay surface into polysiloxanes by a treatment with 3-(triethoxysilyl)propylisocyanate or an azide or propargyl equivalent via the so-called “Click” chemistry [118,119]. 

## 10. CO_2_ Capture for Further Applications and Storage

Once captured, purified, and concentrated, CO_2_ may be compressed, commercialized, and re-injected through adequate technologies into food beverages, refrigerants, fire extinguishers, concrete production, pyrometallurgy, and even in greenhouses and transportation circuits [120]. CO_2_ can also be regarded as a valuable and low-cost material through conversion into added-value chemical products which are mostly carbon-based chemicals and materials such as methane, urea, carbonates, acrylates, epoxy compounds, polymers, drugs, and others [121].

In the current economic context, the conversion of CO_2_ into methane is probably the most interesting way to partially regenerate energy [122]. However, in the near future, even if this so-called methanation process is quite exothermic, it requires molecular hydrogen, specific catalysts, and thermal energy. This reduces the efficiency of its energy balance, unless it is carried out through biological fermentation, like methanization [123]. So far, ample literature has been reported in this regard.

Direct CO_2_ hydrogenation can also generate methanol while Fischer-Tropsch CO_2_ conversion allows for synthesizing chemicals, light olefins, dimethyl ether, liquid fuels, and alcohols [10,124,125]. Here, C–C bond generation should take place after CO_2_ reduction into carbon monoxide or methanol in processes that are still subject to intensive studies aiming for the synthesis of performant catalysts [10,126,127,128]. Tandem catalysis through judicious combination of Fe-based catalysts, metal oxides, and zeolites for offering at least two different types of catalytic sites is a privileged option for this purpose [129,130].

However, these strategies still remain subjects of controversy given that they involve energy consumptions equivalent to CO_2_ release higher than the amount of captured gas. Coupling CO_2_ conversion, for instance through electrochemical processes, with CO_2_ capture may induce a synergy that raises the energy efficiency. This involves the mere suppression of the gas transport and storage and adsorbent regeneration steps more particularly when dealing with CO_2_ chemical capture [131]. In spite of such improvement attempts, large-scale implementation of these valorisation routes still remains limited by low green hydrogen production and high energy consumption [132]. 

In all cases, CO_2_ capture from low CO_2_ atmospheres (ca. 400 ppm) under Direct Air Capture (DAC) conditions is not a viable strategy without consecutive concentration using highly porous adsorbents exhibiting appreciable affinity towards CO_2_ [133]. Base-like solid adsorbents operating via repetitive adsorption-desorption cycles allow for overcoming this shortcoming [41]. Clay–polymer composites have also shown interesting functional properties for this purpose [43,134,135,136], more particularly for environmental applications in atmosphere, waters, and soils [137,138].

Some of these composites are organo-montmorillonites obtained by clay intercalation with organic species displaying organophilic/hydrophilic interactions that promote both dispersion in aqueous media and affinity with the clay surface [43,135], ion exchange with cationic surfactants such as quaternary ammonium [135,139], or others for varied purposes [135,140,141,142,143]. Organo-clays may combine clay minerals such as hydrotalcite [144], montmorillonite [145,146,147], attapulgite [148], and others [43] with natural or synthetic polymers [145,146] or dendrimers [44,50] and others.

Such adsorbents are suitable for CO_2_ capture and concentration for permanent sequestration in underground geological layers, more particularly when operating through the so-called pressure swing (PSA), electric swing (ESA), or temperature swing (TSA) methods [149,150,151]. However, to be sustainable, CO_2_ capture, storage, and utilization (CSU) should also involve technologies that consume less CO_2_-equivalent energy or produce less CO_2_ than the adsorbed amounts. 

## 11. Potential CO_2_ Adsorbents

Selection of CO_2_ adsorbents should also take into account their thermal stability as a constraining factor for the temperature range of the targeted adsorption/desorption process [152]. Carbon-based adsorbents, zeolites, metal oxides, metal organic frameworks, hyper-cross-linked polymers (HCPs), covalent organic frameworks (COFs), conjugated microporous polymers (CMPs), covalent triazine-based frameworks (CTFs), and porous silica with or without modifications were already regarded as prospective CO_2_ adsorbents [153,154,155]. When properly functionalized, organic compounds generally exhibit stronger interaction with CO_2_ [156]. The stability of their interaction with CO_2_ seems to be favoured by increasing amounts in CO_2_ molecules, and they are promoted more by double-bonded carbon atoms as compared to sulphur [156].

Carbonaceous materials such as activated carbons, coal-derived carbons, polymer-derived carbons, metal-organic frameworks-derived carbons, carbon nanotubes, graphene oxides, and carbon aerogels were also found to act as CO_2_ adsorbents [120,157]. Such materials are widely available, highly porous, and thermally stable in O_2_-free atmospheres, being characterized by low production costs and easy synthesis procedures and scaling-up, appreciable affinity towards CO_2_ adsorption under suitable conditions, controllable porosity, and chemical stability. They are not pollutant unlike their synthesis process and can be easily modified by metals or chemical functions [120]. In this regard, low-cost pyrogenic carbons, activated carbons (AC), metal-carbon composites, metal-organic frameworks (MOF), and other carbon nanomaterials already turned out to be quite effective for post-combustion CO_2_ capture [158].

CO_2_ capture has been achieved using metal organic frameworks (MOFs), zeolite imidazolate frameworks (ZIFs), grafted and impregnated polyamines, activated alumina, carbonized porous aromatic frameworks (PAFs), covalent organic frameworks (COFs), porous organic polymers (POPs), mesoporous silica, carbon nanotubes, ionic liquids, phosphates, zeolites, and other molecular sieves [120]. Among the potential CO_2_ adsorbents, ionic liquids appear as probably the less promising materials due to their high operational costs, viscosity, and capacity to promote device corrosion [120] in spite of their higher energy efficiency in DAC-CO_2_ capture compared to alkali and amines as assessed by theoretical calculations alkali and amine [159]. Notwithstanding their unavoidable need for energy-consuming regeneration [160], amines supported by polymers or dendrimers could give rise to promising CO_2_ adsorbents provided that judicious procedures are applied to simultaneously incorporate higher density of adsorption sites and reduce CO_2_ retention strength.

Judicious modifications of COFs and MOFs brought significant improvement in their intrinsic affinity towards CO_2_. Indeed, incorporation of aliphatic amine into covalent organic frameworks (COFs) gave rise to a porous material with promising performances in direct CO_2_ capture from air [161]. Moreover, in porous organic polymers, polyethyleneimine incorporation even improved CO_2_ capture at higher temperature [155]. IRMOF-74-III-CH_2_NH_2_ resulting from primary amine incorporation showed effectiveness in the selective capture of CO_2_ in 65% relative humidity via chemical formation of carbamic acid species, affording appreciable CO_2_ uptake without structure alteration [162]. Metal incorporation is another route for designing MOF-based CO_2_ adsorbents, since finely dispersed particles of Zn-containing MOF turned out to be fairly effective in this gas capture [163]. Cyclodextrin-containing MOF-2 already showed highly selective CO_2_ capture via a weak carbonate-like association that can be easily broken at ambient conditions [164]. However, in spite of their high porosity and adsorption surface, MOFs display a series of shortcomings not only due to potential structural collapse upon vacuum treatments, contact with acidic gases, and thermal instability during regeneration, but also to expensive synthesis procedures [120]. Moreover, most MOFs are synthesized from non-renewable materials, often in harmful solvents originating from fossil sources [164]. 

Aluminosilicates and more particularly zeolites and cationic clay minerals are also interesting materials with potential applications in CO_2_ capture. Zeolites display various selectivity’s in CO_2_ capture, storage, and utilization (CCSU), more particularly in the separation of CO_2_/CH_4_ (biogas) and CO_2_/N_2_ (flue gas) mixtures, according to their framework type, Si/Al ratio, and exchangeable cations [165]. Zeolites, various other molecular sieves, and silica gels also exhibited appreciable CO_2_ retention capacities at relatively low pressures. Nonetheless, the latter were found to progressively decrease in the presence of moisture most likely due to restricted and rigid porosity that unavoidably leads to channel obturation by adsorbed molecules [120]. Even through silicas are known to display unfavourable acidic surfaces for CO_2_ capture, chitosan/mesoporous silica composites (SBA-15 and MCM) were found to act as CO_2_ adsorbents [166].

Various cationic clay minerals such as goethite, hematite, gibbsite, kaolinite, illite, vermiculite, montmorillonite, saponite, nontronite, or even Martian minerals and their anionic counterparts such as LDH and hydrotalcites were already found to exhibit affinity towards CO_2_ [167,168,169]. In hydrotalcite-based sorbents, changes in the charge-compensating anion and/or incorporation of other metals than Mg and Al were found to play key roles in the CO_2_ retention capacity (CRC) [170,171]. Mixed Mg and Al oxides, produced by LDH alteration, showed promising prospects for CO_2_ capture of industrial flue emissions [172]. Nonetheless, metal oxides and clay materials display only low porosity and adsorption surface in their native state [43].

Incidentally, unlike expanded material structures obtained through sophisticated synthesis procedures, polyol-intercalated montmorillonites can be conveniently prepared from widely available and low-cost clay minerals combined with commercial or natural OH-organic compounds [42]. More convenient and eco-friendly adsorbents for the reversible capture of CO_2_ are certainly polyglycerol-montmorillonites obtained by using soya oil as a polyglycerol source [41]. Polyol dendrimers insertion in LDHs resulted in hybrid surfaces with optimum affinity due to an attenuation of the relatively strong base character of the starting LDH material [35]. It is worth quoting that polyol-modified Mg-Al LDH produced higher CRC values at lower temperature, even higher than with LDH-supported amine [35]. This was explained in terms of attenuated basicity and improved physical adsorption at the expense of chemosorption.

As a common feature, OH-enriched clay minerals display optimum adsorptive properties resulting from a judicious compromise between the highest adsorption capacities and the lowest desorption temperatures [41]. This should be the main criterion for the selection of future CO_2_ adsorbents, taking into account that the latter should at least display Bronsted basicity, bearing electron pair donor atoms, being capable of promoting Lewis acid-base interactions and/or H-bridges.

## 12. CO_2_ Retention Capacity and Parameter Effects

Given the wide variety of CO_2_ adsorbents tested so far, a comparison of the CRC values of only some adsorbent family representatives allows for stating that expanded structures and *N*-group incorporation afford the highest CO_2_ uptakes (Table 2). The CRC of carbonaceous materials such as activated carbon, carbon nanofibers, hollow carbon spheres, and biochar are strongly dependent on micropore volume and surface area which, in turn, are determined by their carbonization and activation temperature and time and presence of moisture. Their chemical modification by *N*-containing functional groups was found to enhance their adsorption capacities [173]. Chemical grafting of amine into organic frameworks (COFs) resulted in a marked increase in CO_2_ by a factor of more than one thousand, which was much more pronounced in the presence of water molecules [161]. This beneficial effect of *N*-compound incorporation was also noticed with Polyethylenimine (PEI)-impregnated fumed silica. Such composites turned out to fairly effective in CO_2_ capture under DAC even in very diluted CO_2_ atmospheres [174].

The CO_2_ adsorption capacity from the air of carbon-based materials was found to increase with increasing porosity and specific surface area (SSA) [185]. The highest CO_2_ uptakes were registered for biochar, activated biochar, and to a lesser extent activated carbons. Here, the specific surface area (SSA) accounts for the accessible adsorption surface and is probably one of the most important factors that determine the CO_2_ uptake.

For instance, on Zn-containing MOF, CO_2_ exchange speed appears to be favoured by decreasing particle size, i.e., by increasing contact surface [163]. This is why it is worth emphasizing that the very CRC value is not relevant when not reported to the specific surface area, and no CRC comparison between different adsorbents can be accurate and reliable. On the basis of the so-called surface efficiency factor (SEF) [42] or surface affinity factor (SAF) [99] defined as the CRC/SSA ratio, it appears that, in spite of their apparently low CRC value, but with SSA barely reaching 60 m^2^/g, Boltorn^TM^ polyol-montmorillonite composites display much higher SAF values of more than 2.73 µmol CO_2_/m^2^ as compared to amine-based adsorbents and highly sophisticated MOF structures. 

Amine incorporation in highly porous silica often results in a marked decay of the specific surface area from more than 700 m^2^/g to less than 30 m^2^/g that reduces the adsorption site accessibility [180,181]. ZVM incorporation also induced a depletion of the accessibility towards the terminal Si-OH groups as illustrated by a noticeable CRC decay in spite of an additional basicity, presumably involving metal-carbonate association [186]. Therefore, the use of costly expanded structures only for hosting amine or OH-compounds is questionable, and interest has to be focused on low amine loading at least for reducing CO_2_ retention strength without shading the intrinsic basicity of the host-surface or affecting its accessibility. A judicious strategy should arise from a rigorous compromise between high and low CO_2_ retention capacity and strength in correlation with the CRC direct proportionality with both the porosity and adsorption site density. 

Incorporation of zero-valent metal (ZVM) in mesoporous silica was also found to induce only slight CO_2_ improvements. The latter are progressively annihilated upon repeated adsorption/desorption cycles and are not revived after alternate rehydration. This result is of great importance because it clearly demonstrates the following: *i*. ZVM weakly contribute to CO_2_ most likely via the formation of carbonates; *ii*. High temperature regeneration leads to framework alteration, presumably as a result of the decreasing amount of silanol groups upon irreversible dehydroxylation [182,183].

This additional affinity towards CO_2_ induced by ZVM insertion was also observed in organo-bentonite-supported Cu^0^ and Pd^0^ particles intended for the reversible capture of hydrogen at ambient conditions. Metal-organo-clays were also found to display interesting affinity towards CO_2_ [98]. However, the improvements still remain weak by far as compared to optimum incorporation of OH-functionalized organic moiety that prevents structure compaction upon excessive H-bridges. Therefore, in the absence of diffusion hindrance, all polyol-intercalated montmorillonite displayed CO_2_ adsorption capacity of up to 11.70–16.42 µmol.g^−1^, but with much lower desorption temperatures with full regeneration at 35–40 °C, or at 20 °C upon forced convection or in the presence of KOH pills [184].

LDH-polyol composites were also found to display sufficiently weak basicity to exert only physical interactions towards more than one CO_2_ molecule [35]. These interactions are so weak that any variation of temperature and carrier gas throughput can result in marked fluctuations of the CO_2_ uptake. This explains their easy regeneration. As already stated, OH incorporation favours their basicity attenuation at the expense of the CRC, even though the latter increased by about two times after intercalation with Boltorn^TM^ dendrimers H20, H30, and H20, even reaching values higher than those reported for some amine-containing LDH [187]. The CO_2_ retention strength is expected to be governed not only by the electronic structure, polarity, and porosity of the adsorbent [188], but also by the air chemical composition in terms of air components, moisture, and pollutants [120].

## 13. Hydroxyl Affinity towards CO_2_ and Water

The hydroxyl groups and the basicity of a solid surface are expected to play key roles in CO_2_ capture by solid surfaces, and more particularly on clay minerals. For instance, anionic clay minerals display stronger basicity affording higher CRC levels as compared to their cationic counterparts such as montmorillonite, where a weak Lewis basicity arises from the oxygen atoms surrounding the ion-exchange sites [184,189]. On such surfaces, the in-plane silanols are expected to exhibit amphoteric to slightly basic character [52]. Their OH groups were already found to act as adsorption sites for CO_2_ molecules [190,191,192,193].

In clay-supported OH- compounds, the affinity towards water and CO_2_ were found to increase almost linearly with the increasing number of OH groups, as long as this site still remains accessible [43]. For instance, the highest CRC values were registered only for variable optimum amounts of Boltorn^TM^ dendrimer according to the molecular structure [40]. Excessive density of inserted OH groups beyond a certain threshold appears to produce excessive H-bridges and a dramatic CRC decrease due to entanglement compaction. Moreover, on OH-enriched adsorbents, it appears that more than one CO_2_ molecule absorbs onto each OH group, presumably due a multilayer retention [192,193]. Thus, the CO_2_/OH ratio was not only higher than unity but also increased with increasing CO_2_ concentration in the impregnating atmosphere [186]. CO_2_ retention strength is so weak that the gas can be totally released at 35–40 °C, or 20 °C upon forced convection, or even in CO_2_-free media in the presence of more basic species such as NaOH/KOH pills [40]. The occurrence of purely weak physical -HO:CO_2_ interaction was demonstrated by shifts in IR bands assigned to the bending vibration of CO_2_ and asymmetric stretching of the C=O bonds [40] and of solid-state ^13^C NMR signals for cyclodextrin-containing MOF-2. The latter was explained in terms of CO_2_ capture via the reversible formation of a weak carbonate-like association [164].

In Boltorn^TM^ H30-modified montmorillonite, the slight shift in the TPD desorption peak from 68 °C to 73 °C upon dendrimer insertion provides clear evidence of an enhancement in CO_2_ retention strength. This is due to the slightly higher basicity of the hydroxyl groups as compared to the clay silanol Si–OH groups. This basicity is an additional contribution to that of the lattice oxygen of the silica silanols [194]. The role of both silanol and incorporated OH groups was already confirmed by the noticeable CRC decay after metal incorporation as assessed by CO_2_-TPD measurements between 20 to 80 °C [194]. This was explained by the rise in competitive -HO:Metal interaction at the expense of CO_2_ upon metal incorporation. Reportedly, CO_2_ capture by activated carbon derived from longan seeds involves strong electrostatic -HO:CO_2_ interaction which is involved between the OH group and CO_2_. Not excessive iron insertion at an optimum concentration of about 1% on the surface produced the highest CO_2_ adsorption capacity [175].

The occurrence of CO_2_ interaction with OH groups should also involve CO_2_-H_2_O interdependence. Here, the very hydrophilic character of the hydroxyl groups induced by the incorporation of OH-dendrimers suggests a contribution of the adsorbed water to CO_2_ capture [40]. This was already illustrated by an almost linearly proportional CRC increase with increasing moisture content when the incorporated hydroxyls are accessible to both CO_2_ and water molecules. On mesoporous silica, ZMV incorporation was found not only to reduce the CRC but also an attenuation of water retention strength, most likely due to a competitive -HO:Metal interaction at the expense of CO_2_ and water molecules [186]. The role of moisture is expected to differ from that of the exchangeable cation hydration. Indeed, Fe^2+-^exchange montmorillonite was found to display lower affinity towards CO_2_ in spite of higher hydrophilic character as compared to its Na^+^-exchanged counterpart [195]. This was explained by a higher polarizing power of Fe^2+^ cation and stronger Bronsted acidity arising from the dissociation of water molecules surrounding hydrated Fe^2+^ cation ([Fe(H_2_O)_x_]^2+^→[Fe(H_2_O)_x−1_.OH]^+^ + H^+^) [184].

## 14. Metal-Carbonate Association

ZVM incorporation in mesoporous silicas was found to affect CRC with multiple possible effects: *i.* Porosity decrease due to the formation of bulky and bare MNP with low dispersion in the absence of dispersing agent; *ii.* The rise of competitive -HO:Metal interaction; *ii.* Simultaneous metal chelation in the form of metal-carbonate association [186]. This was already demonstrated by severe thermal treatment that unavoidably leads to silanol depletion via irreversible dehydroxylation [182,183]. Modified SBA-15 showed different Metal:CO_2_ interactions illustrated by two TPD signals at 140–230 °C and 300–540 °C attributed to bidentate and unidentate carbonates, respectively. Signal intensity comparison suggests a preponderance of unidentate carbonates [182]. Both types of carbonates were also predicted by calculations and ^13^C-NMR on the Mg−O sites of mixed Mg and Al oxides close to an Al atom [172]. The formation of carbonate on metal oxides was confirmed by other calculations which revealed that surface hydroxyls act as effective sites for CO_2_ adsorption via hydrogen bonds during CO_2_ electroreduction reactions [196].

Amine-based adsorbents are also known to interact with CO_2_ through the formation of carbonates, whose stability was found to vary according to the modification procedures of the very amine moiety. Reportedly, in functionalized amines, the grafting of electron withdrawing groups generates less stable CO_2_ reaction products and could be regarded as being more suitable for lower energy regeneration as compared to those produced by electron-donating groups [197]. Here, it appears that the stabilization of the resulting bicarbonates is due to the capacity of the amine to promote hydrogen bonding [197]. This result is of great importance because it provides clear evidence that the CO_2_ retention strength could be attenuated and opens promising prospects for the modification of commercially available polyamido-amine dendrimers (PAMAM).

Click chemical grafting of glucoside moieties on organobentonite resulted in Gluco-organobentonites (GOB) and insertion of metal nanoparticle (MNP) induced an almost total depletion of the hydrophilic character and CO_2_ retention capacity [186]. The fact that no total suppression of the affinity toward CO_2_ was noticed can be explained by the formation of metal carbonate complexes via competitive interactions of the terminal OH groups with the incorporated MNPs. Simultaneously, DSC measurements in dry helium, between 20 and 80 °C, revealed a significant decrease in the CO_2_ retention strength expressed in terms of a decrease in the CO_2_ desorption heat from 10.5 cal/g for bentonite and 12.5 cal/g for Gluco-organobentonite down to 5.25 for Cu°-GOB and 4.5 cal/g for Pd°-GOB [98]. This result is of great interest because it clearly demonstrates that metal incorporation could be a viable strategy for low regeneration energy.

## 15. Conclusions

The main fallout of this literature synthesis and analysis resides in the possibility to build theoretical bases for designing novel adsorbents for truly reversible CO_2_. This can be achieved by using low-cost raw materials as inorganic supports that already possess expanded 2-D (clay) or 3-D (zeolite) frameworks. Various organic moieties commercially available or synthesized bearing terminal hydroxyls, thiolated polyol, amino, phosphino, phosphoryl, and other groups can be inserted in clay minerals (e.g., montmorillonite) and zeolites. Amino and hydroxyl groups can be optimally combined with incorporated zero-valent metal or oxides to prepare CO_2_ adsorbents affording simultaneously high CO_2_ uptake and low retention strength for energy regeneration in CO_2_-free enclosures, without heating or forced convection. The literature provides valuable findings that can be systematically analysed and converted into criteria for designing the targeted CO_2_ adsorbents. Towards this goal, dendrimers bearing amino groups such as PAMAMs that can be adequately modified appear as the most promising candidates.

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
