# Peer review of "Innovative Strategy for Truly Reversible Capture of Polluting Gases—Application to Carbon Dioxide"

_ijms, 2023, doi:10.3390/ijms242216463_

Round 1
Reviewer 1 Report
Comments and Suggestions for Authors
The paper is a highly significant review, focusing on a topic that has gained considerable attention in recent years. It requires some improvements before it can be published in this journal:
- The introduction of the work is quite comprehensive. I even have questions whether the introduction should be shortened, as sometimes it repeats information.
- In the page 4 of the manuscript, there is a typo of English, 'et.' I presume it should be 'and.'
- I believe the paper needs an additional section discussing different types of CO2 conversion into products of interest for society. For instance, the conversion of CO2 into methane, among other applications, should be addressed.
- The recently developed and tested hydrotalcites should also be discussed (see examples below, among others):
https://doi.org/10.1016/j.seppur.2019.03.026
https://doi.org/10.1016/j.seppur.2019.116140
Author Response
Manuscript ID: ijms-2699643
Type of manuscript: Review
Title: Innovative Strategy for Truly Reversible Capture of Polluting Gases-
Application to Carbon Dioxide
Authors: Abdelkrim AZZOUZ *, René Roy
Corresponding author: Abdelkrim Azzouz
Reviewer #1:
Reviewer’s general comments
The paper is a highly significant review, focusing on a topic that has gained considerable attention in recent years. It requires some improvements before it can be published in this journal:
Authors’ response
Done. The manuscript has been fully revised taking into account all comments and suggestions as written by each reviewer (without indexes and superscripts in many words). Changes have been marked in green color in their corresponding location in the text. A file with point-by-point responses to all reviewers requests has been attached to the revised manuscript.
Reviewer’s specific comments
- There were some very basic grammatical errors in the paper that needed grammatical revision.
Authors’ response
Done. A full spelling of the text has been performed, which allowed removing some typos and basic grammatical errors.
Reviewer’s specific comments
- The introduction of the work is quite comprehensive. I even have questions whether the introduction should be shortened, as sometimes it repeats information.
Authors’ response
Done. Some repeated information has been suppressed and some sentences have been rephrased for improving the conciseness.
Reviewer’s specific comments
- In the page 4 of the manuscript, there is a typo of English, 'et.' I presume it should be 'and.'
Authors’ response
Done. This mistake has been addressed. The change made is marked in green color.
Reviewer’s specific comments
- I believe the paper needs an additional section discussing different types of CO2 conversion into products of interest for society. For instance, the conversion of CO2 into methane, among other applications, should be addressed.
Authors’ response
Done. Many statements and a new 2nd paragraph have been incorporated in section 10. CO2 capture for further applications and storage, in page 14 of this revised manuscript. This entire section title has also been thoroughly rewritten accordingly.
Reviewer’s specific comments
- The recently developed and tested hydrotalcites should also be discussed (see examples below, among others):
https://doi.org/10.1016/j.seppur.2019.03.026
https://doi.org/10.1016/j.seppur.2019.116140
Authors’ response
Done. The suggested references have been added into a discussion. See changes marked in green color in the last three paragraphs of section 11. Potential CO2 adsorbents, in page 16 of the revised manuscript.

Reviewer 2 Report
Comments and Suggestions for Authors
The authors in the present review manuscript to show that the deep analysis and data comparison of the main strategies under-taken for achieving truly reversible capture of carbon dioxide involving optimized gas uptakes while affording weakest retention strength. So far, most strategies failed because the estimated amount of CO2 produced by equivalent energy was higher than that captured. A more viable and sustainable approach in the present context of persistent fossil fuel-dependent economy should be based on a judicious compromise between effective CO2 capture with lowest energy for adsorbent regeneration. The most relevant example is that of so-called promising technologies based on amino adsorbents which unavoidably require thermal regeneration. In contrast, OH-functionalized adsorbents barely reach satisfactory CO2 uptakes but act a breathing adsorbent affording easy gas release even under ambient conditions or in CO2-free atmospheres. Between these two opposite approaches, there should exist smart approaches to tailor CO2 retention strength even at the expense of the gas uptake. Among these, incorporation of zero-valent metal and/or OH-enriched amines or amine-enriched polyol species are probably the most promising. The main findings provided by literature are herein deeply and systematically analyzed for highlighting the main criteria that allow designing ideal CO2 adsorbent properties. The authors should address the following issues and information’s before publication acceptance in the prestigious ‘International Journal of Molecular Sciences’ Journal:
1. In the review paper, authors should add a Table that compares different CO2 removal/capture methods, parameters, removal capacity and properties? Authors may also add the advantages and disadvantages of each CO2 removal method.
2. Author may incorporate a graph that shows how many papers are published each year on CO2 capture?
3. In Introduction, authors should mention the CO2 concentration increment each year?
4. Authors can incorporate paragraph on research gaps and challenges for future recommendations.
5. In Introduction, authors need to support some sentences with references for example, “fossil fuel combustions predominantly eject CO2, sulfur oxides (SOx, mainly SO2 and SO3), while agricultural and derived processes produce high amount of nitogen oxides (NOx, mainly nitrate, nitrite and nitrogen protoxide) beside others.” Authors may go through these publications and refer accordingly: https://doi.org/10.1007/s11665-018-3192-2 & https://doi.org/10.1021/ef040059h

Comments on the Quality of English LanguageMinor editing of English language required.
Author Response
Manuscript ID: ijms-2699643
Type of manuscript: Review
Title: Innovative Strategy for Truly Reversible Capture of Polluting Gases-
Application to Carbon Dioxide
Authors: Abdelkrim AZZOUZ *, René Roy
Corresponding author: Abdelkrim Azzouz
Reviewer #2:
Reviewer’s general comments
The authors in the present review manuscript to show that the deep analysis and data comparison …. for highlighting the main criteria that allow designing ideal CO2 adsorbent properties. The authors should address the following issues and information’s before publication acceptance in the prestigious ‘International Journal of Molecular Sciences’ Journal:
Authors’ response
Done. The manuscript has been fully revised taking into account all comments and suggestions as written by each reviewer (without indexes and superscripts in many words). Changes have been marked in green color in their corresponding location in the text. A file with point-by-point responses to all reviewers requests has been attached to the revised manuscript.
Reviewer’s specific comments
- In the review paper, authors should add a Table that compares different CO2 removal/capture methods, parameters, removal capacity and properties? Authors may also add the advantages and disadvantages of each CO2 removal method.
Authors’ response
Done. Assuming that the reviewer is referring to the type of flue emissions to be treated, a new table, many statements and some paragraphs have been added to the revised manuscript. See changes marked green color in section 4. Strategies for CO2 capture and potential valorization routes, in pages 4-6.
Reviewer’s specific comments
- Author may incorporate a graph that shows how many papers are published each year on CO2 capture?
Authors’ response
Done. This is an excellent suggestion. However, given that such graphs have been systematically reviewed (see added reference), we decided to just indicate the appropriate reference number by adding the following statement in the first paragraph of section 4. Strategies for CO2 capture, in page 4 of the revised manuscript
‘’ For tackling this issue, scientists have triggered extensive research, more particularly after 2000, inasmuch as between the period 2001-2021, 18,500 publications CO2 capture and sequestration-related publications have been reported (ref Yu et al.) with an additional 4,500 in the last two years only’’ .
The following reference has been added as well: Yu, X.; Catanescu, C. O.; Bird, R. E.; Satagopan, S.; Baum, Z. J.; Lotti Diaz, L. M.; Zhou, Q. A. Trends in Research and Development for CO2 Capture and Sequestration. ACS Omega 2023, 8, 11643−11664.
Reviewer’s specific comments
- In Introduction, authors should mention the CO2 concentration increment each year?
Authors’ response
Done. The requested data have been added as new statements marked in green color in the second paragraph of section 3. CO2 natural cycle and major emission sources, after scheme 2 in page 4 of the revised manuscript.
Reviewer’s specific comments
- Authors can incorporate paragraph on research gaps and challenges for future recommendations.
Authors’ response
Done. Two paragraphs have been added at the end of section 3. CO2 natural cycle and major emission sources, in page 4 of the revised manuscript.
Reviewer’s specific comments
- In Introduction, authors need to support some sentences with references for example, “fossil fuel combustions predominantly eject CO2, sulfur oxides (SOx, mainly SO2 and SO3), while agricultural and derived processes produce high amount of nitogen oxides (NOx, mainly nitrate, nitrite and nitrogen protoxide) beside others.” Authors may go through these publications and refer accordingly: https://doi.org/10.1007/s11665-018-3192-2 & https://doi.org/10.1021/ef040059h
Authors’ response
The first publication suggested (https://doi.org/10.1007/s11665-018-3192-2) deals with Room Temperature Adsorptive Removal of Thiophene over Zinc Oxide-Based Adsorbents and has nothing to do with the present work.
The second publication suggested (https://doi.org/10.1021/ef040059h) deals with the adsorptive properties of five zeolites. The paper concluded that ‘’the CO2 adsorption capacity was significantly lower at 120 °C than at ambient temperature’’ and that high CO2 adsorption capacity are favored by high pressure. Both statements are well-known and already reported by numerous works including ours, which were already cited in the present manuscript.
However, in order to avoid controversy, the second reference suggested has been incorporated in a discussion. See changes marked in green color at the end of the last paragraph of section 6. CO2 Adsorption on solids, in page 6 of the revised manuscript.
Reviewer’s specific comments (Comments on the Quality of English Language)
Minor editing of English language required.
Authors’ response
Done. A full spelling of the text has been performed, which allowed removing some typos and basic grammatical errors.

Round 2
Reviewer 1 Report
Comments and Suggestions for Authors
-